# STCN: Stochastic Temporal Convolutional Networks

**Emre Aksan & Otmar Hilliges**
Department of Computer Science
ETH Zurich, Switzerland
{emre.aksan, otmar.hilliges}@inf.ethz.ch

## Abstract

Convolutional architectures have recently been shown to be competitive on many sequence modelling tasks when compared to the de-facto standard of recurrent neural networks (RNNs), while providing computational and modeling advantages due to inherent parallelism. However, currently there remains a performance gap to more expressive stochastic RNN variants, especially those with several layers of dependent random variables. In this work, we propose stochastic temporal convolutional networks (STCNs), a novel architecture that combines the computational advantages of temporal convolutional networks (TCN) with the representational power and robustness of stochastic latent spaces. In particular, we propose a hierarchy of stochastic latent variables that captures temporal dependencies at different time-scales. The architecture is modular and flexible due to decoupling of deterministic and stochastic layers. We show that the proposed architecture achieves state of the art log-likelihoods across several tasks. Finally, the model is capable of predicting high-quality synthetic samples over a long-range temporal horizon in modeling of handwritten text.

## 1 Introduction

Generative modeling of sequence data requires capturing long-term dependencies and learning of correlations between output variables at the same time-step. Recurrent neural networks (RNNs) and its variants have been very successful in a vast number of problem domains which rely on sequential data. Recent work in audio synthesis, language modeling and machine translation tasks (Dauphin et al., 2016; Van Den Oord et al., 2016; Dieleman et al., 2018; Gehring et al., 2017) has demonstrated that temporal convolutional networks (TCNs) can also achieve at least competitive performance without relying on recurrence, and hence reducing the computational cost for training.

Both RNNs and TCNs model the joint probability distribution over sequences by decomposing the distribution over discrete time-steps. In other words, such models are trained to predict the next step, given all previous time-steps. RNNs are able to model long-term dependencies by propagating information through their deterministic hidden state, acting as an internal memory. In contrast, TCNs leverage large receptive fields by stacking many dilated convolutions, allowing them to model even longer time scales up to the entire sequence length. It is noteworthy that there is no explicit temporal dependency between the model outputs and hence the computations can be performed in parallel. The TCN architecture also introduces a temporal hierarchy: the upper layers have access to longer input sub-sequences and learn representations at a larger time scale. The local information from the lower layers is propagated through the hierarchy by means of residual and skip connections (Van Den Oord et al., 2016; Bai et al., 2018).

However, while TCN architectures have been shown to perform similar or better than standard recurrent architectures on particular tasks (Van Den Oord et al., 2016; Bai et al., 2018), there currently remains a performance gap to more recent stochastic RNN variants (Bayer & Osendorfer, 2014; Chung et al., 2015; Fabius & van Amersfoort, 2014; Fraccaro et al., 2016; Goyal et al., 2017; Shabanian et al., 2017). Following a similar approach to stochastic RNNs, Lai et al. (2018) present a significant improvement in the log-likelihood when a TCN model is coupled with latent variables, albeit at the cost of limited receptive field size.

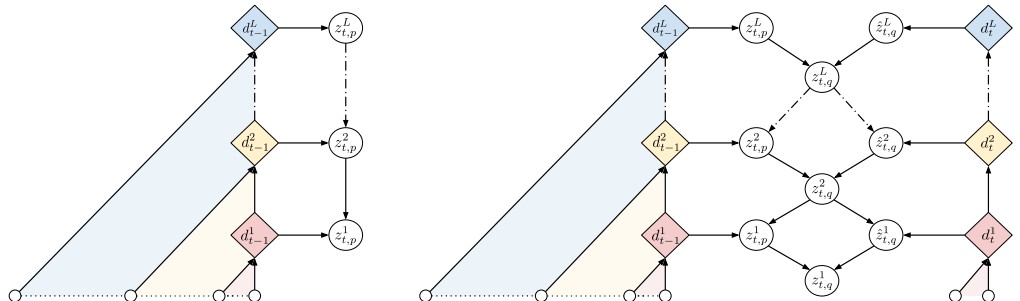

**Figure 1:** The computational graph of generative (left) and inference (right) models of STCN. The approximate posterior $q$ is conditioned on $\mathbf{d}_t$ and is updated by the prior $p$ which is conditioned on the TCN representations of the previous time-step $\mathbf{d}_{t-1}$. The random latent variables at the upper layers have access to a long history while lower layers receive inputs from more recent time steps.

In this work we propose a new approach for augmenting TCNs with random latent variables, that decouples deterministic and stochastic structures yet leverages the increased modeling capacity efficiently. Motivated by the simplicity and computational advantages of TCNs and the robustness and performance of stochastic RNNs, we introduce stochastic temporal convolutional networks (STCN) by incorporating a hierarchy of stochastic latent variables into TCNs which enables learning of representations at many timescales. However, due to the absence of an internal state in TCNs, introducing latent random variables analogously to stochastic RNNs is not feasible. Furthermore, defining conditional random variables across time-steps would result in breaking the parallelism of TCNs and is hence undesirable.

In STCN the latent random variables are arranged in correspondence to the temporal hierarchy of the TCN blocks, effectively distributing them over the various timescales (see figure 1). Crucially, our hierarchical latent structure is designed to be a modular add-on for any temporal convolutional network architecture. Separating the deterministic and stochastic layers allows us to build STCNs without requiring modifications to the base TCN architecture, and hence retains the scalability of TCNs with respect to the receptive field. This conditioning of the latent random variables via different timescales is especially effective in the case of TCNs. We show this experimentally by replacing the TCN layers with stacked LSTM cells, leading to reduced performance compared to STCN.

We propose two different inference networks. In the canonical configuration, samples from each latent variable are passed down from layer to layer and only one sample from the lowest layer is used to condition the prediction of the output. In the second configuration, called STCN-dense, we take inspiration from recent CNN architectures (Huang et al., 2017) and utilize samples from all latent random variables via concatenation before computing the final prediction.

Our contributions can thus be summarized as: 1) We present a modular and scalable approach to augment temporal convolutional network models with effective stochastic latent variables. 2) We empirically show that the STCN-dense design prevents the model from ignoring latent variables in the upper layers (Zhao et al., 2017). 3) We achieve state-of-the-art log-likelihood performance, measured by ELBO, on the IAM-OnDB, Deepwriting, TIMIT and the Blizzard datasets. 4) Finally we show that the quality of the synthetic samples matches the significant quantitative improvements.

## 2 BACKGROUND

Auto-regressive models such as RNNs and TCNs factorize the joint probability of a variable-length sequence $\mathbf{x} = \{x_1, \ldots, x_T\}$ as a product of conditionals as follows:

$$p_\theta(\mathbf{x}) = \prod_{t=1}^{T} p_\theta(x_t | x_{1:t-1}) \quad , \tag{1}$$

where the joint distribution is parametrized by $\theta$. The prediction at each time-step is conditioned on all previous observations. The observation model is frequently chosen to be a Gaussian or Gaussian mixture model (GMM) for real-valued data, and a categorical distribution for discrete-valued data.

## 2.1 TEMPORAL CONVOLUTIONAL NETWORKS

In TCNs the joint probabilities in Eq. (1) are parametrized by a stack of convolutional layers. *Causal convolutions* are the central building block of such models and are designed to be asymmetric such that the model has no access to future information. In order to produce outputs of the same size as the input, zero-padding is applied at every layer.

In the absence of a state transition function, a large receptive field is crucial in capturing long-range dependencies. To avoid the need for vast numbers of causal convolution layers, typically *dilated* convolutions are used. Exponentially increasing the dilation factor results in an exponential growth of the receptive field size with depth (Yu & Koltun, 2015; Van Den Oord et al., 2016; Bai et al., 2018). In this work, without loss of generality, we use the building blocks of Wavenet (Van Den Oord et al., 2016) as gated activation units (van den Oord et al., 2016) have been reported to perform better.

A deterministic TCN representation $d_t^l$ at time-step $t$ and layer $l$ summarizes the input sequence $x_{1:t}$:

$$d_t^l = \text{Conv}^{(l)}(d_t^{l-1}, d_{t-j}^{l-1}) \quad \text{and} \quad d_t^1 = \text{Conv}^{(1)}(x_t, x_{t-j}) \quad , \tag{2}$$

where the filter width is 2 and $j$ denotes the dilation step. In our work, the stochastic variables $z^l, l = 1 \dots L$ are conditioned on TCN representations $d^l$ that are constructed by stacking $K$ Wavenet blocks over the previous $d^{l-1}$ (for details see Figure 4 in Appendix).

## 2.2 NON-SEQUENTIAL LATENT VARIABLE MODELS

VAEs (Kingma & Welling, 2013; Rezende et al., 2014) introduce a latent random variable $\mathbf{z}$ to learn the variations in the observed non-sequential data where the generation of the sample $\mathbf{x}$ is conditioned on the latent variable $\mathbf{z}$. The joint probability distribution is defined as:

$$p_\theta(\mathbf{x}, \mathbf{z}) = p_\theta(\mathbf{x}|\mathbf{z})p_\theta(\mathbf{z}) \quad , \tag{3}$$

and parametrized by $\theta$. Optimizing the marginal likelihood is intractable due to the non-linear mappings between $\mathbf{z}$ and $\mathbf{x}$ and the integration over $\mathbf{z}$. Instead the VAE framework introduces an approximate posterior $q_\phi(\mathbf{z}|\mathbf{x})$ and optimizes a lower-bound on the marginal likelihood:

$$\log p_\theta(\mathbf{x}) \geq -KL(q_\phi(\mathbf{z}|\mathbf{x})||p_\theta(\mathbf{z})) + \mathbb{E}_{q_\phi(\mathbf{z}|\mathbf{x})}[\log p_\theta(\mathbf{x}|\mathbf{z})] \quad , \tag{4}$$

where $KL$ denotes the Kullback-Leibler divergence. Typically the prior $p_\theta(\mathbf{z})$ and the approximate $q_\phi(\mathbf{z}|\mathbf{x})$ are chosen to be in simple parametric form, such as a Gaussian distribution with diagonal covariance, which allows for an analytical calculation of the $KL$-term in Eq. (4).

## 2.3 STOCHASTIC RNNS

An RNN captures temporal dependencies by recursively processing each input, while updating an internal state $h_t$ at each time-step via its state-transition function:

$$h_t = f^{(h)}(x_t, h_{t-1}) \quad , \tag{5}$$

where $f^{(h)}$ is a deterministic transition function such as LSTM (Hochreiter & Schmidhuber, 1997) or GRU (Cho et al., 2014) cells. The computation has to be sequential because $h_t$ depends on $h_{t-1}$.

The VAE framework has been extended for sequential data, where a latent variable $z_t$ augments the RNN state $h_t$ at each sequence step. The joint distribution $p_\theta(\mathbf{x}, \mathbf{z})$ is modeled via an auto-regressive model which results in the following factorization:

$$p_\theta(\mathbf{x}, \mathbf{z}) = \prod_{t=1}^{T} p_\theta(x_t|z_{1:t}, x_{1:t-1})p_\theta(z_t|x_{1:t-1}, z_{1:t-1}) \quad . \tag{6}$$

In contrast to the fixed prior of VAEs, $\mathcal{N}(\mathbf{0}, \mathbf{I})$, sequential variants define prior distributions conditioned on the RNN hidden state $\mathbf{h}$ and implicitly on the input sequence $\mathbf{x}$ (Chung et al., 2015).

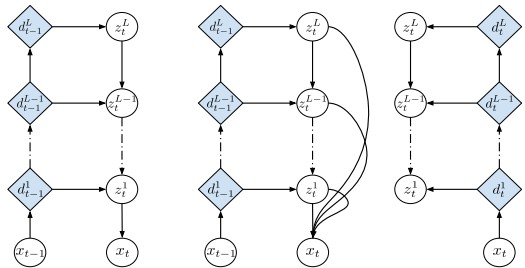

**Figure 2:** Graphical model view of generative models of STCN (*left*) and STCN-dense (*middle*), and the inference model (*right*), which is shared by both variants. Diamonds represent the outputs of deterministic dilated convolution blocks where the dependence of $d_t$ on the past inputs is not shown for clarity (see Eq. (2)). $x_t$ and $z_t$ are observable inputs and latent random variables, respectively. The generative task is to predict the next step in the sequence, given all past steps. Note that in the STCN-dense variant the next step is conditioned on all latent variables $z_t^l$ for $l = 1 \ldots L$.

## 3 STOCHASTIC TEMPORAL CONVOLUTIONAL NETWORKS

The mechanics of STCNs are related to those of VRNNs and LVAEs. Intuitively, the RNN state $h_t$ is replaced by temporally independent TCN layers $d_t^l$. In the absence of an internal state, we define hierarchical latent variables $z_t^l$ that are conditioned *vertically*, i.e., in the same time-step, but independent *horizontally*, i.e., across time-steps. We follow a similar approach to LVAEs (Sønderby et al., 2016) in defining the hierarchy in a *top-down* fashion and in how we estimate the approximate posterior. The inference network first computes the approximate likelihood, and then this estimate is corrected by the prior, resulting in the approximate posterior. The TCN layers **d** are shared between the inference and generator networks, analogous to VRNNs (Chung et al., 2015).

Figure 2 depicts the proposed STCN as a graphical model. STCNs consist of two main modules: the deterministic temporal convolutional network and the stochastic latent variable hierarchy. For a given input sequence $\mathbf{x} = \{x_t\}, t = 1 \ldots T$ we first apply dilated convolutions over the entire sequence to compute a set of deterministic representations $d_t^l, l = 1 \ldots L$. Here, $d_t^l$ corresponds to the output of a block of dilated convolutions at layer $l$ and time-step $t$. The output $d_t^l$ is then used to update a set of random latent variables $z_t^l$ arranged to correspond with different time-scales.

To preserve the *parallelism* of TCNs, we do not introduce an explicit dependency between different time-steps. However, we suggest that conditioning a latent variable $z_t^{l-1}$ on the preceding variable $z_t^l$ implicitly introduces temporal dependencies. Importantly, the random latent variables in the upper layer have access to a larger receptive field due to its deterministic input $d_{t-1}^l$, whereas latent random variables in lower layers are updated with different, more local information. However, the latent variable $z_t^{l-1}$ may receive longer-range information from $z_t^l$.

The generative and inference models are jointly trained by optimizing a step-wise variational lower bound on the log-likelihood (Kingma & Welling, 2013; Rezende et al., 2014). In the following sections we describe these components and build up the lower-bound for a single time-step $t$.

### 3.1 GENERATIVE MODEL

Each sequence step $x_t$ is generated from a set of latent variables $z_t$, split into layers as follows:

$$p_\theta(z_t|x_{1:t-1}) = p_\theta(z_t^L|d_{t-1}^L) \prod_{l=1}^{L-1} p_\theta(z_t^l|z_t^{l+1}, d_{t-1}^l) \quad , \qquad (7)$$

$$\text{where} \quad p_\theta(z_t^l|z_t^{l+1}, d_{t-1}^l) = \mathcal{N}(\mu_{t,p}^l, \sigma_{t,p}^l) \quad \text{and} \quad [\mu_{t,p}^l, \sigma_{t,p}^l] = f_p^{(l)}(z_t^{l+1}, d_{t-1}^l) \quad . \qquad (8)$$

Here the prior is modeled by a Gaussian distribution with diagonal covariance, as is common in the VAE framework. The subscript $p$ denotes items of the generative distribution. For the inference distribution we use the subscript $q$. The distributions are parameterized by a neural network $f_p^{(l)}$ and conditioned on: (1) the $d_{t-1}^l$ computed by the dilated convolutions from the previous time-step, and

(2) a sample from the preceding level at the same time-step $z_t^{l+1}$. Please note that at inference time we draw samples from the approximate posterior distribution $z_t^{l+1} \sim q_\phi(z_t^{l+1}|\cdot)$. The generative model, on the other hand, uses the prior $z_t^{l+1} \sim p_\theta(z_t^{l+1}|\cdot)$.

We propose two variants of the observation model. In the non-sequential scenario, the observations are defined to be conditioned on only last latent variable in the hierarchy, i.e., $p_\theta(x_t|z_t^1)$, following Sønderby et al. (2016); Gulrajani et al. (2016) and Rezende et al. (2014) our STCN variant uses the same observation model, allowing for an efficient optimization. However, latent units are likely to become inactive during training in this configuration (Burda et al., 2015; Bowman et al., 2015; Zhao et al., 2017) resulting in a loss of representational power.

The latent variables at different layers are conditioned on different contexts due to the inputs $d_t^l$. Hence, the latent variables are expected to capture complementary aspects of the temporal context. To propagate the information all the way to the final prediction and to ensure that gradients flow through all layers, we take inspiration from Huang et al. (2017) and directly condition the output probability on samples from *all* latent variables. We call this variant of our architecture *STCN-dense*.

The final predictions are then computed by the respective observation functions:

$$p_\theta(x_t|z_t) = f^{(o)}(z_t^1) \quad \text{and} \quad p_\theta^{dense}(x_t|z_t) = f^{(o)}(z_t^1, z_t^2 \ldots z_t^L) \quad , \tag{9}$$

where $f^{(o)}$ corresponds to the output layer constructed by stacking 1D convolutions or Wavenet blocks depending on the dataset.

## 3.2 INFERENCE MODEL

In the original VAE framework the inference model is defined as a bottom-up process, where the latent variables are conditioned on the stochastic layer below. Furthermore, the parameterization of the prior and approximate posterior distributions are computed separately (Burda et al., 2015; Rezende et al., 2014). In contrast, Sønderby et al. (2016) propose a top-down dependency structure shared across the generative and inference models. From a probabilistic point of view, the approximate Gaussian likelihood, computed bottom-up by the inference model, is combined with the Gaussian prior, computed top-down from the generative model. We follow a similar procedure in computing the approximate posterior.

First, the parameters of the approximate likelihood are computed for each stochastic layer $l$:

$$[\hat{\mu}_{t,q}^l, \hat{\sigma}_{t,q}^l] = f_q^{(l)}(z_t^{l+1}, d_t^l) \quad , \tag{10}$$

followed by the downward pass, recursively computing the prior and approximate posterior by precision-weighted addition:

$$\sigma_{t,q}^l = \frac{1}{(\hat{\sigma}_{t,q}^l)^{-2} + (\sigma_{t,p}^l)^{-2}} \quad ,$$
$$\mu_{t,q}^l = \sigma_{t,q}^l(\hat{\mu}_{t,q}^l(\hat{\sigma}_{t,q}^l)^{-2} + \mu_{t,p}^l(\sigma_{t,p}^l)^{-2}) \quad . \tag{11}$$

Finally, the approximate posterior has the same decomposition as the prior (see Eq. (7)):

$$q_\phi(z_t|x_{1:t}) = q_\phi(z_t^L|d_t^L) \prod_{l=1}^{L-1} q_\phi(z_t^l|z_t^{l+1}, d_t^l) \quad , \tag{12}$$

$$q_\phi(z_t^l|z_t^{l+1}, d_t^l) = \mathcal{N}(\mu_{t,q}^l, \sigma_{t,q}^l) \quad . \tag{13}$$

Note that the inference and generative network share the parameters of dilated convolutions $\text{Conv}^{(l)}$.

## 3.3 LEARNING

The variational lower-bound on the log-likelihood at time-step $t$ can be defined as follows:

$$\log p(x_t) \geq \mathbb{E}_{q_\phi(z_t|x_t)}[\log p_\theta(x_t|z_t)] - D_{KL}(q_\phi(z_t|x_{1:t})||p_\theta(z_t|x_{1:t-1}))$$
$$= \mathbb{E}_{q_\phi(z_t^1 \ldots z_t^L|x_t)}[\log p_\theta(x_t|z_t^1 \ldots z_t^L)] - D_{KL}(q_\phi(z_t^1 \ldots z_t^L|x_{1:t})||p_\theta(z_t^1 \ldots z_t^L|x_{1:t-1}))$$
$$\mathcal{L}_t(\theta, \phi; x_t) = \mathcal{L}_t^{Recon} + \mathcal{L}_t^{KL}. \tag{14}$$

Using the decompositions from Eq. (7) and (12), the Kullback-Leibler divergence term becomes:

$$\mathcal{L}_t^{KL} = - D_{KL}(q_\phi(z_t^L|d_t^L)||p_\theta(z_t^L|d_{t-1}^L))$$
$$- \sum_{l=1}^{L-1} \mathbb{E}_{q_\phi(z_t^{l+1}|\cdot)}[D_{KL}(q_\phi(z_t^l|z_t^{l+1},d_t^l)||p_\theta(z_t^l|z_t^{l+1},d_{t-1}^l))] \quad . \tag{15}$$

The KL term is the same for the STCN and STCN-dense variants. The reconstruction term $\mathcal{L}_t^{Recon}$, however, is different. In STCN we only use samples from the lowest layer of the hierarchy, whereas in STCN-dense we use all latent samples in the observation model:

$$\mathcal{L}_t^{Recon} = \mathbb{E}_{q_\phi(z_t^1...z_t^L|x_t)}[\log p_\theta(x_t|z_t^1)] \quad , \tag{16}$$

$$\mathcal{L}_t^{Recon-dense} = \mathbb{E}_{q_\phi(z_t^1...z_t^L|x_t)}[\log p_\theta(x_t|z_t^1 \ldots z_t^L)] \quad . \tag{17}$$

In the dense variant, samples drawn from the latent variables $z_t^l$ are carried over the dense connections. Similar to Maaløe et al. (2016), the expectation over $z_t^l$ variables are computed by Monte Carlo sampling using the reparameterization trick (Kingma & Welling, 2013; Rezende et al., 2014).

Please note that the computation of $\mathcal{L}_t^{Recon-dense}$ does not introduce any additional computational cost. In STCN, all latent variables have to be visited in terms of ancestral sampling in order to draw the latent sample $z_t^1$ for the observation $x_t$. Similarly in STCN-dense, the same intermediate samples $z_t^l$ are used in the prediction of $x_t$.

One alternative option to use the latent samples could be to sum individual samples before feeding them into the observation model, i.e., $sum([z_t^1 \ldots z_t^L])$, (Maaløe et al., 2016). We empirically found that this does not work well in STCN-dense. Instead, we concatenate all samples $[z_t^1 \circ \cdots \circ z_t^L]$ analogously to DenseNet (Huang et al., 2017) and (Kaiser et al., 2018).

## 4 EXPERIMENTS

We evaluate the proposed variants STCN and STCN-dense both quantitatively and qualitatively on modeling of digital handwritten text and speech. We compare with vanilla TCNs, RNNs, VRNNs and state-of-the art models on the corresponding tasks.

In our experiments we use two variants of the Wavenet model: (1) the original model proposed in (Van Den Oord et al., 2016) and (2) a variant that we augment with skip connections analogously to STCN-dense. This additional baseline evaluates the benefit of learning *multi-scale* representations in the deterministic setting. Details of the experimental setup are provided in the Appendix. Our code is available at `https://ait.ethz.ch/projects/2019/stcn/`.

**Handwritten text:** The IAM-OnDB and Deepwriting datasets consist of digital handwriting sequences where each time-step contains real-valued $(x, y)$ pen coordinates and a binary *pen-up* event. The IAM-OnDB data is split and pre-processed as done in (Chung et al., 2015). Aksan et al. (2018) extend this dataset with additional samples and better pre-processing.

Table 1 reveals that again both our variants outperform the vanilla variants of TCNs and RNNs on IAM-OnDB. While the stochastic VRNN and SWaveNet are competitive wrt to the STCN variant, both are outperformed by

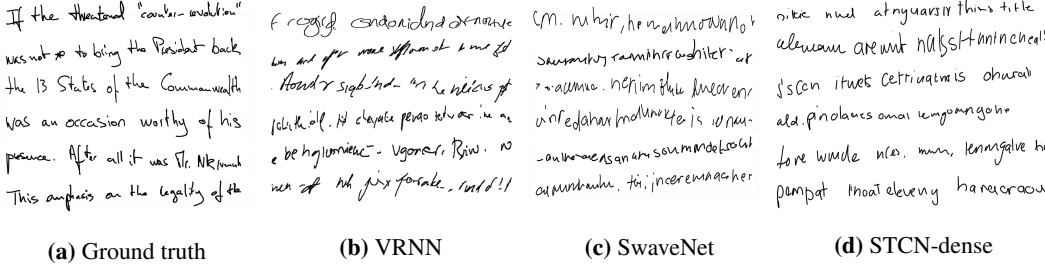

| **(a)** Ground truth | **(b)** VRNN | **(c)** SwaveNet | **(d)** STCN-dense |

**Figure 3:** (a) Handwriting samples from IAM-OnDB dataset. Generated samples from (b) VRNN, (c) SWaveNet and (d) our model STCN-dense. Each line corresponds to one sample.

**Table 1:** Average log-likelihood per sequence on TIMIT, Blizzard, IAM-OnDB and Deepwriting datasets. (Normal) and (GMM) stand for unimodal Gaussian or multi-modal Gaussian Mixture Model (GMM) as the observation model (Graves, 2013; Chung et al., 2015). Asterisks * indicate that we used our re-implementation only for the Deepwriting dataset.

| Models | TIMIT | Blizzard | IAM-OnDB | Deepwriting |
|---|---|---|---|---|
| Wavenet (GMM) | 30188 | 8190 | 1381 | 612 |
| Wavenet-dense (GMM) | 30636 | 8212 | 1380 | 642 |
| RNN (GMM) [Chung et al. (2015)] | 26643 | 7413 | 1358 | 528 * |
| VRNN (Normal) [Chung et al. (2015)] | $\approx 30235$ | $\approx 9516$ | $\approx 1354$ | $\geq 495$ * |
| VRNN (GMM) [Chung et al. (2015)] | $\approx 29604$ | $\approx 9392$ | $\approx 1384$ | $\geq 673$ * |
| SRNN (Normal) [Fraccaro et al. (2016)] | $\geq 60550$ | $\geq 11991$ | n/a | n/a |
| Z-forcing (Normal) [Goyal et al. (2017)] | $\geq 70469$ | $\geq 15430$ | n/a | n/a |
| Var. Bi-LSTM (Normal) [Shabanian et al. (2017)] | $\geq 73976$ | $\geq 17319$ | n/a | n/a |
| SWaveNet (Normal) [Lai et al. (2018)] | $\geq 72463$ | $\geq 15708$ | $\geq 1301$ | n/a |
| STCN (GMM) | $\geq 69195$ | $\geq 15800$ | $\geq 1338$ | $\geq 605$ |
| STCN-dense (GMM) | $\geq 71386$ | $\geq 16288$ | $\geq \mathbf{1796}$ | $\geq \mathbf{797}$ |
| STCN-dense-large (GMM) | $\geq \mathbf{77438}$ | $\geq \mathbf{17670}$ | n/a | n/a |

the STCN-dense version. The same relative ordering is maintained on the Deepwriting dataset, indicating that the proposed architecture is robust across datasets.

Fig. 3 compares generated handwriting samples. While all models produce consistent style, our model generates more natural looking samples. Note that the spacing between words is clearly visible and most of the letters are distinguishable.

**Speech modeling:** TIMIT and Blizzard are standard benchmark dataset in speech modeling. The models are trained and tested on 200 dimensional real-valued amplitudes. We apply the same pre-processing as Chung et al. (2015). For this task we introduce STCN-dense-large, with increased model capacity. Here we use 512 instead of 256 convolution filters. Note that the total number of model parameters is comparable to SWaveNet and other SOA models.

On TIMIT, STCN-dense (Table 1) significantly outperforms the vanilla TCN and RNN, and stochastic models. On the Blizzard dataset, our model is marginally better than the Variational Bi-LSTM. Note that the inference models of SRNN (Fraccaro et al., 2016), Z-forcing (Goyal et al., 2017), and Variational Bi-LSTM (Shabanian et al., 2017) receive future information by using backward RNN cells. Similarly, SWaveNet (Lai et al., 2018) applies causal convolutions in the backward direction. Hence, the latent variable can be expected to model future dynamics of the sequence. In contrast, our models have only access to information up to the current time-step. These results indicate that the STCN variants perform very well on the speech modeling task.

**Latent Space Analysis:** Zhao et al. (2017) observe that in hierarchical latent variable models the upper layers have a tendency to become inactive, indicated by a low KL loss (Sønderby et al., 2016; Dieng et al., 2018). Table 2 shows the KL loss per latent variable and the corresponding log-likelihood measured by ELBO in our models. Across the datasets it can be observed that our models make use of many of the latent variables which may explain the strong performance across tasks in terms of log-likelihoods. Note that STCN uses a standard hierarchical structure. However, individual latent variables have different information context due to the corresponding TCN block's receptive field. This observation suggests that the proposed combination of TCNs and stochastic variables is indeed effective. Furthermore, in STCN we see a similar utilization pattern of the $z$ variables across tasks, whereas STCN-dense may have more flexibility in modeling the temporal dependencies within the data due to its dense connections to the output layer.

**Table 2:** KL-loss per latent variable computed over the entire test split. KL5 corresponds to the KL-loss of the top-most latent variable.

| Dataset (Model) | ELBO | KL | KL1 | KL2 | KL3 | KL4 | KL5 |
|---|---|---|---|---|---|---|---|
| IAM-OnDB (STCN-dense) | $\geq 1796.3$ | 1653.9 | 17.9 | 1287.4 | 305.3 | 41.0 | 2.4 |
| IAM-OnDB (STCN) | $\geq 1339.2$ | 964.2 | 846.0 | 105.2 | 12.9 | 0.1 | 0.0 |
| TIMIT (STCN-dense) | $\geq 71385.9$ | 22297.5 | 16113.0 | 5641.6 | 529.0 | 8.3 | 5.7 |
| TIMIT (STCN) | $\geq 69194.9$ | 23118.3 | 22275.5 | 487.2 | 355.5 | 0.0 | 0.0 |

**Replacing TCN with RNN:** To better understand potential symergies between dilated CNNs and the proposed latent variable hierarchy, we perform an ablation study, isolating the effect of TCNs and the latent space. To this end the deterministic TCN blocks are replaced with LSTM cells by keeping the latent structure intact. We dub this condition LadderRNN. We use the TIMIT and IAM-OnDB datasets for evaluation. Table 3 summarizes performance measured by the ELBO.

The most direct translation of the the STCN architecture into an RNN counterpart has 25 stacked LSTM cells with 256 units each. Similar to STCN, we use 5 stochastic layers (see Appendix 7.1). Note that stacking this many LSTM cells is unusual and resulted in instabilities during training. Hence, the performance is similar to vanilla RNNs. The second LadderRNN configuration uses 5 stacked LSTM cells with 512 units and a one-to-one mapping with the stochastic layers. On the TIMIT dataset, all LadderRNN configurations show a significant improvement. We also observe a pattern of improvement with densely connected latent variables.

This experiments shows that the proposed modular latent variable design does allow for the usage of different building blocks. Even when attached to LSTM cells, it boosts the log-likelihood performance (see 5x512-LadderRNN), in particular when used with dense connections. However, the empirical results suggest that the densely connected latent hierarchy interacts particularly well with dilated CNNs. We suggest this is due to the hierarchical nature on both sides of the architecture. On both datasets STCN models achieved the best performance and significantly improve with dense connections. This supports our contribution of a latent variable hierarchy, which models different aspects of information from the input time-series.

**Table 3:** ELBO of LadderRNN and STCN models using the same latent space configuration. The prefix of a model entries denote the number of RNN or TCN layers and unit size per layer. Models have similar number of trainable parameters.

| Models | TIMIT | IAM-OnDB |
|---|---|---|
| 25x256-LadderRNN (Normal) | $\geq 28207$ | $\geq 1305$ |
| 25x256-LadderRNN-dense (Normal) | $\geq 27413$ | $\geq 1278$ |
| 25x256-LadderRNN (GMM) | $\geq 24839$ | $\geq 1381$ |
| 25x256-LadderRNN-dense (GMM) | $\geq 26240$ | $\geq 1377$ |
| 5x512-LadderRNN (Normal) | $\geq 49770$ | $\geq 1299$ |
| 5x512-LadderRNN-dense (Normal) | $\geq 48612$ | $\geq 1374$ |
| 5x512-LadderRNN (GMM) | $\geq 47179$ | $\geq 1359$ |
| 5x512-LadderRNN-dense (GMM) | $\geq 50113$ | $\geq 1581$ |
| 25x256-STCN (Normal) | $\geq 64913$ | $\geq 1327$ |
| 25x256-STCN-dense (Normal) | $\geq 70294$ | $\geq 1729$ |
| 25x256-STCN (GMM) | $\geq 69195$ | $\geq 1339$ |
| 25x256-STCN-dense (GMM) | $\geq \mathbf{71386}$ | $\geq \mathbf{1796}$ |

## 5 RELATED WORK

Rezende et al. (2014) propose Deep Latent Gaussian Models (DLGM) and Sønderby et al. (2016) propose the Ladder Variational Autoencoder (LVAE). In both models the latent variables are hierarchically defined and conditioned on the preceding stochastic layer. LVAEs improve upon DLGMs via implementation of a top-down hierarchy both in the generative and inference model. The approximate posterior is computed via a precision-weighted update of the approximate likelihood (i.e., the inference model) and prior (i.e., the generative model). Similarly, the PixelVAE (Gulrajani et al., 2016) incorporates a hierarchical latent space decomposition and uses an autoregressive decoder. Zhao et al. (2017) show under mild conditions that straightforward stacking of latent variables (as is done e.g. in LVAE and PixelVAE) can be ineffective, because the latent variables that are not directly conditioned on the observation variable become inactive.

Due to the nature of the sequential problem domain, our approach differs in the crucial aspects that STCNs use dynamic, i.e., conditional, priors (Chung et al., 2015) at every level. Moreover, the hierarchy is not only implicitly defined by the network architecture but also explicitly defined by the information content, i.e., receptive field size. Dieng et al. (2018) both theoretically and empirically show that using skip connections from the latent variable to every layer of the decoder increases mutual information between the latent and observation variables. Similar to Dieng et al. (2018) in STCN-dense, we introduce skip connections from all latent variables to the output. In STCN the model is expected to encode and propagate the information through its hierarchy.

Yang et al. (2017) suggest using autoregressive TCN decoders to remedy the posterior collapse problem observed in language modeling with LSTM decoders (Bowman et al., 2015). van den Oord et al. (2017) and Dieleman et al. (2018) use TCN decoders conditioned on discrete latent variables to model audio signals.

Stochastic RNN architectures mostly vary in the way they employ the latent variable and parametrize the approximate posterior for variational inference. Chung et al. (2015) and Bayer & Osendorfer (2014) use the latent random variable to capture high-level information causing the variability observed in sequential data. Particularly Chung et al. (2015) shows that using a conditional prior rather than a standard Gaussian distribution is very effective in sequence modeling. In (Fraccaro et al., 2016; Goyal et al., 2017; Shabanian et al., 2017), the inference model, i.e., the approximate posterior, receives both the past and future summaries of the sequence from the hidden states of forward and backward RNN cells. The KL-divergence term in the objective enforces the model to learn predictive latent variables in order to capture the future states of the sequence.

Lai et al. (2018)'s SWaveNet is most closely related to ours. SWaveNet also introduces latent variables into TCNs. However, in SWaveNet the deterministic and stochastic units are coupled which may prevent stacking of larger numbers of TCN blocks. Since the number of stacked dilated convolutions determines the receptive field size, this directly correlates with the model capacity. For example, the performance of SWaveNet on the IAM-OnDB dataset degrades after stacking more than 3 stochastic layers (Lai et al., 2018), limiting the model to a small receptive field. In contrast, we aim to preserve the flexibility of stacking dilated convolutions in the base TCN. In STCNs, the deterministic TCN units do not have any dependency on the stochastic variables (see Figure 1) and the ratio of stochastic to deterministic units can be adjusted, depending on the task.

## 6 CONCLUSION

In this paper we proposed STCNs, a novel auto-regressive model, combining the computational benefits of convolutional architectures and expressiveness of hierarchical stochastic latent spaces. We have shown the effectivness of the approach across several sequence modelling tasks and datasets. The proposed models are trained via optimization of the ELBO objective. Tighter lower bounds such as IWAE (Burda et al., 2015) or FIVO (Maddison et al., 2017) may further improve modeling performance. We leave this for future work.

## ACKNOWLEDGEMENTS

This work was supported in parts by the ERC grant OPTINT (StG-2016-717054). We gratefully acknowledge the support of NVIDIA Corporation with the donation of the Titan Xp GPU used for this research.

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

## 7 APPENDIX

### 7.1 NETWORK DETAILS

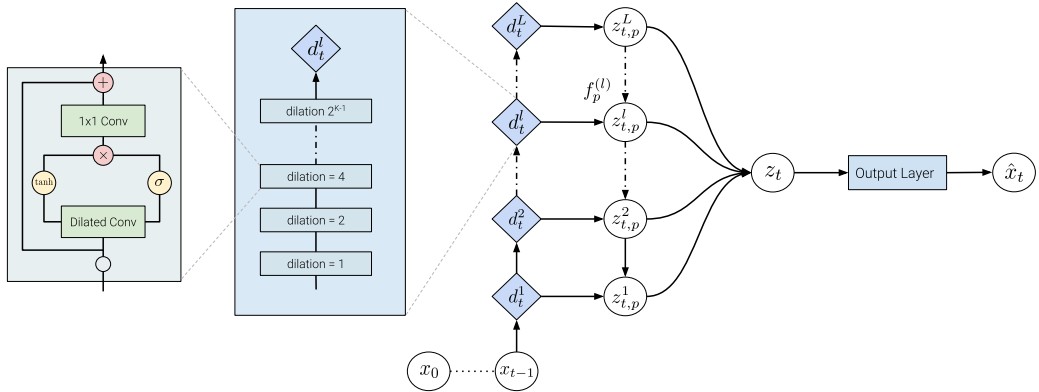

**Figure 4:** Generative model of STCN-dense architecture. Building blocks are highlighted. Note that the dependence of $d_t^l, l = 1 \cdots L$ on past inputs is not visualized for clarity.

The network architecture of the proposed model is illustrated in Fig. 4. We make only a small modification to the vanilla Wavenet architecture. Instead of using skip connections from Wavenet blocks, we only use the latent sample $z_t$ in order to make a prediction of $x_t$. In STCN-dense configuration, $z_t$ is the concatenation of all latent variables in the hierarchy, i.e., $z_t = [z_t^1 \circ \cdots \circ z_t^L]$, whereas in STCN only $z_t^1$ is fed to the output layer.

Each stochastic latent variable $z_t^l$ (except the top-most $z_t^L$) is conditioned on a deterministic TCN representation $d_t^l$ and the preceding random variable $z_t^{l+1}$. The latent variables are calculated by using the latent layers $f_p^{(l)}$ or $f_q^{(l)}$ which are neural networks.

We do not define a latent variable per TCN layer. Instead, the stochastic layers are uniformly distributed where each random variable is conditioned on a number of stacked TCN layers $d_t^l$. We stack $K$ Wavenet blocks (see figure 4 left) with exponentially increasing dilation size.

**Observation Model:** We use Normal or GMM distributions with 20 components to model real-valued data. All Gaussian distributions have diagonal covariance matrix.

**Output layer $f^{(o)}$:** For the IAM-OnDB and Deepwriting datasets we use 1D convolutions with ReLU nonlinearity. We stack 5 of these layers with 256 filters and filter size 1.

For TIMIT and Blizzard datasets Wavenet blocks in the output layer perform significantly better. We stack 5 Wavenet blocks with dilation size 1. For each convolution operation in the block we use 256 filters. The filter size of the dilated convolution is set to 2. The STCN-dense-large model is constructed by using 512 filters instead of 256.

**TCN blocks $d_t^l$:** The number of Wavenet blocks is usually determined by the desired receptive field size.

- For the handwriting datasets $K = 6$ and $L = 5$. In total we have 30 Wavenet blocks where each convolution operation has 256 filters with size 2.

- For speech datasets $K = 5$ and $L = 5$. In total we have 25 Wavenet blocks where each convolution operation has 256 filters with size 2. The large model configuration uses 512 filters.

**Latent layers $f_p^{(l)}$ and $f_q^{(l)}$:** The number of stochastic layers per task is given by $L$. We used $[32, 16, 8, 5, 2]$ dimensional latent variables for the handwriting tasks. It is $[256, 128, 64, 32, 16]$ for speech datasets. Note that the first entry of the list corresponds to $z^1$.

The mean and sigma parameters of the Normal distributions modeling the latent variables are calculated by the $f_p^{(l)}$ and $f_q^{(l)}$ networks. We stack 2 1D convolutions with ReLU nonlinearity and filter size 1. The number of filters are the same as the number of Wavenet block filters for the corresponding task.

Finally, we clamped the latent sigma predictions between 0.001 and 5.

## 7.2 TRAINING DETAILS

In all STCN experiments we applied KL annealing. In all tasks, the weight of the KL term is initialized with $0$ and increased by $1 \times e^{-4}$ at every step until it reaches $1$.

The batch size was 20 for all datasets except for Blizzard where it was 128.

We use the ADAM optimizer with its default parameters and exponentially decay the learning rate. For the handwriting datasets the learning rate was initialized with $5 \times e^{-4}$ and followed a decay rate of $0.94$ over $1000$ decay steps. On the speech datasets it was initialized with $1 \times e^{-3}$ and decayed with a rate of $0.98$. We applied early stopping by measuring the ELBO performance on the validation splits.

We implement STCN models in Tensorflow (Abadi et al., 2016). Our code and models achieving the SOA results are available at `https://ait.ethz.ch/projects/2019/stcn/`.

## 7.3 DETAILED RESULTS

Here we provide the extended results table with Normal observation model entries for available models.

**Table 4:** Average $\log$-likelihood per sequence on TIMIT, Blizzard, IAM-OnDB and Deepwriting datasets. (Normal) and (GMM) stand for unimodal Gaussian or multi-modal Gaussian Mixture Model (GMM) as the observation model (Graves, 2013; Chung et al., 2015). Asterisks $^*$ indicate that we used our re-implementation only for the Deepwriting dataset.

| Models | TIMIT | Blizzard | IAM-OnDB | Deepwriting |
|---|---|---|---|---|
| Wavenet (Normal) | -7443 | 3784 | 1053 | 337 |
| Wavenet (GMM) | 30188 | 8190 | 1381 | 612 |
| Wavenet-dense (Normal) | -8579 | 3712 | 1030 | 323 |
| Wavenet-dense (GMM) | 30636 | 8212 | 1380 | 642 |
| RNN (Normal) [Chung et al. (2015)] | -1900 | 3539 | 1016 | 363 * |
| RNN (GMM) [Chung et al. (2015)] | 26643 | 7413 | 1358 | 528 * |
| VRNN (Normal)[Chung et al. (2015)] | $\approx 30235$ | $\approx 9516$ | $\approx 1354$ | $\geq 495$ * |
| VRNN (GMM) [Chung et al. (2015)] | $\approx 29604$ | $\approx 9392$ | $\approx 1384$ | $\geq 673$ * |
| SRNN (Normal) [Fraccaro et al. (2016)] | $\geq 60550$ | $\geq 11991$ | n/a | n/a |
| Z-forcing (Normal)[Goyal et al. (2017)] | $\geq 70469$ | $\geq 15430$ | n/a | n/a |
| Var. Bi-LSTM (Normal)[Shabanian et al. (2017)] | $\geq 73976$ | $\geq 17319$ | n/a | n/a |
| SWaveNet (Normal)[Lai et al. (2018)] | $\geq 72463$ | $\geq 15708$ | $\geq 1301$ | n/a |
| STCN(Normal) | $\geq 64913$ | $\geq 13273$ | $\geq 1327$ | $\geq 575$ |
| STCN(GMM) | $\geq 69195$ | $\geq 15800$ | $\geq 1338$ | $\geq 605$ |
| STCN-dense(Normal) | $\geq 70294$ | $\geq 15950$ | $\geq 1729$ | $\geq 740$ |
| STCN-dense(GMM) | $\geq 71386$ | $\geq 16288$ | $\geq \mathbf{1796}$ | $\geq \mathbf{797}$ |
| STCN-dense-large (GMM) | $\geq \mathbf{77438}$ | $\geq \mathbf{17670}$ | n/a | n/a |

