# OpenReview forum: "STCN: Stochastic Temporal Convolutional Networks"
_ICLR.cc/2019/Conference_

### Official Review · AnonReviewer3 · 2018-11-02
**Clearly written, but lacking comparisons**

**Rating:** 6
**Confidence:** 4

**Review:**

The focus on novelty (mentioned in both the abstract, and conclusion as a direct claim) in the presentation hurts the paper overall. Without stronger comparison to other closely related work, and lack of citation to several closely related models, the claim of novelty isn't defined well enough to be useful. Describing what parts of this model are novel compared to e.g. Stochastic WaveNet or the conditional dilated convolutional decoder of "Improved VAE for Text ..." (linked below, among many others) would help strengthen the novelty claim, if the claim of novelty is needed or useful at all. Stochastic WaveNet in particular seems very closely related to this work, as does PixelVAE. In addition, use of autoregressive models conditioned on (non-variational, in some sense) latents have been shown in both VQ-VAE and ADA among others, so a discussion would help clarify the novelty claim.

Empirical results are strong, though (related to the novelty issue) there should be greater comparison both quantitatively and qualitatively to further work. In particular, many of the papers linked below show better empirical results on the same datasets. Though the results are not always directly comparable, a discussion of *why* would be useful - similar to how Z-forcing was included.

In the qualitative analysis, it would be good to see a more zoomed out view of the text (as in VRNN), since one of the implicit claims of the improvement from dense STCN is improved global coherence by direct connection to the "global latents". As it stands now the text samples are a bit too local to really tell. In addition, the VRNN samples look quite a bit different than what the authors present in their work - what implementation was used for the VRNN samples (they don't appear to be clips from the original paper)?

On the MNIST setting, there are many missing numbers in the table from related references (some included below), and the >= 60.25 number seems so surprising as to be (possibly) incorrect - more in-depth analysis of this particular result is needed. Overall the MNIST result needs more description and relation to other work, for both sequential and non-sequential models.

The writing is well-done overall, and the presented method and diagrams are clear. My primary concern is in relation to related work, clarification of the novelty claim, and more comparison to existing methods in the results tables.

Variational Bi-LSTM https://arxiv.org/abs/1711.05717

Stochastic WaveNet https://arxiv.org/abs/1806.06116

PixelVAE https://arxiv.org/abs/1611.05013

Filtering Variational Objectives https://github.com/tensorflow/models/tree/master/research/fivo

Improved Variational Autoencoders for Text Modeling using Dilated Convolutions https://arxiv.org/abs/1702.08139

Temporal Sigmoid Belief Networks for Sequential Modeling http://papers.nips.cc/paper/5655-deep-temporal-sigmoid-belief-networks-for-sequence-modeling

Neural Discrete Representation Learning (VQ-VAE) https://arxiv.org/abs/1711.00937

The challenge of realistic music generation: modelling raw audio at scale (ADA) https://arxiv.org/abs/1806.10474

Learning hierarchical features from Generative Models https://arxiv.org/abs/1702.08396

Avoiding Latent Variable Collapse with Generative Skip Models https://arxiv.org/abs/1807.04863

EDIT: Updated score after second revisions and author responses

---

> ### Author Response · Authors · 2018-11-17
> **Comments and Clarifications**
>
> ***Missing citations and novelty claim
> We thank the reviewer for useful pointers to additional related papers. In the revised version, we added a more complete related work section. In particular, we discuss the most closely related Stochastic Wavenet paper in detail. While SWaveNet and ours combine TCNs with stochastic variables there are important differences in how this is achieved. Furthermore, we show that these design choices have implications in terms of modelling power and our architecture outperforms SWaveNet despite not having access to future information. Furthermore, we provide log-likelihood results from Variational Bi-LSTM and Stochastic Wavenet are inserted into the result table. In order to provide more evidence, we also include experiments on the Blizzard dataset.
>
> We would like to emphasize that the main difference between our model and the models with autoregressive decoders (i.e., PixelVAE, Improved Variational Autoencoders for Text Modeling using Dilated Convolutions) is the sequential structure of our latent space. For every timestep x_t we have a corresponding latent variable z_t, similar to stochastic RNNs, which helps modeling the uncertainty in sequence data. We aim to combine TCNs with a powerful latent variable structure to better model sequence data rather than learning disentangled or interpretable representations. The updated results show that our design successfully preserves the modeling capacity of TCNs and representation power of latent variables.
>
> *** Handwriting sample figure.
> In order to make a direct comparison, we include a new figure (similar to VRNN) comparing generated handwriting samples of VRNN, Stochastic Wavenet and STCN-dense. The original figure referred to by the reviewer is now in the Appendix.
>
> *** MNIST results
> (Also see the answer to R1) We include a new figure comparing the performance of STCN, STCN-dense and VRNN on single test samples from seq-MNIST. We find that STCN-dense makes very precise probability predictions for the pixel values as opposed to other models, this explains the drastic increase in likelihood performance.
> We include a table providing KL loss per latent variable across the whole dataset. We also provide a comparison between SKIP-VAE (Avoiding Latent Variable Collapse with Generative Skip Models) and our model. It shows that STCN-dense effectively uses the latent space capacity (indicated by high KL values) and encodes the required information to reconstruct the input sequence. We also provide generated MNIST samples in order to show that the discrepancy between the prior and approximate posterior does not degrade generative modeling capacity.
> Finally, in our MNIST experiments, we followed Z-forcing paper’s instructions. See reply to R1 for details of the experimental protocol.

---

> > ### Comment · AnonReviewer3 · 2018-11-28
> > **Updates look pretty good overall, but one huge issue remains.**
> >
> > The new updates are much improved, and the direct discussion of closely related work greatly relieves my concern in this area. Thank you for the updates and improvements.
> >
> > However, I cannot accept the MNIST STCN-dense number without extraordinary evidence (the level of which is frankly impossible to give in a double blind conference review). It would be a serious issue for any follow-on work, and without extremely strong (to the level of replication / rerunning the code and at least some days of digging) evidence, I cannot update my score due to this point alone.
> >
> > I *strongly* urge the authors to avoid this particular number (even leaving the pure STCN without dense connections seems fine), as the rest of the results seem quite solid and the contribution of the paper is meaningful - there is no need to have this controversy when the focus of the paper is not really MNIST modeling. Other papers with similarly radical improvements (~62 to far lower) have had to be withdrawn or reworked due to methodology concerns, and I would really not like to see the same thing here, when it isn't necessary for the message or concept of the paper.
> >
> > As far as debug strategies if you really, really want to be confident in the result, you can multiply every contribution in the dense connections which is connected to the original input by 0 (this may be tricky), the number should fall back to something reasonable. If it breaks entirely, or if the number stays really low, these are both serious causes for concern. Adding huge amounts of noise on these connections should also force the model to fall back to alternate connections, and shouldn't break things utterly if it is a real scenario - it should fall back to something roughly like the standard STCN.
> >
> > Without that particular number as an issue, I would definitely raise my score - the updates address most of my other concerns.

---

> > > ### Author Response · Authors · 2018-12-03
> > > **Removing the MNIST experiment**
> > >
> > > We are glad that the reviewer finds the paper much improved. Furthermore, we agree that the MNIST experiment is not important to convey the contribution of our work and hence we are happy to remove it since it does not add much in this context. Since discarding only the STCN-dense result only, would result in an incomplete experiment, we suggest to remove the whole MNIST experiment - guidance welcome.  We also appreciate the debug suggestions. We will follow up on these.

---

### Official Review · AnonReviewer2 · 2018-11-03
**Interesting new architecture, but some clarity issues**

**Rating:** 6
**Confidence:** 3

**Review:**

This paper introduces a new stochastic neural network architecture for sequence modeling. The model as depicted in figure 2 has a ladder-like sequence of deterministic convolutions bottom-up and stochastic Gaussian units top-down.

I'm afraid I have a handful of questions about aspects of the architecture that I found confusing. I have a difficult time relating my understanding of the architecture described in figure 2 with the architecture shown in figure 1 and the description of the wavenet building blocks. My understanding of wavenet matches what is shown in the left of figure 1: the convolution layers d_t^l depend on the convolutional layers lower-down in the model, thus with each unit d^l having dependence which reaches further and further back in time as l increases. I don't understand how to reconcile this with the computation graph in figure 2, which proposes a model which is Markov! In figure 2, each d_{t-1}^l depends only on on the other d_{t-1} units and the value of x_{t-1}, which then (in the left diagram of figure 2) generate the following x_t, via the z_t^l. Where did the dilated convolutions go…? I thought at first this was just a simplification for the figure, but then in equation (4), there is d_t^l = Conv^{(l)}(d_t^{l-1}). Shouldn't this also depend on d_{t-1}^{l-1}…? or, where does the temporal information otherwise enter at all? The only indication I could find is in equation (13), which has a hidden unit defined as d_t^1 = Conv^{(1)}(x_{1:t}).

Adding to my confusion, perhaps, is the way that the "inference network" and "prior" are described as separate models, but sharing parameters. It seems that, aside from the initial timesteps, there doesn't need to be any particular prior or inference network at all: there is simply a transition model from x_{t-1} to x_{t}, which would correspond to the Markov operator shown in the left and middle sections of figure 2. Why would you ever need the right third of figure 2? This is a model that estimates z_t given x_t. But, aside from at time 0, we already have a value x_{t-1}, and a model which we can use to estimate z_t  given x_{t-1}…!

What are the top-to-bottom functions f^{(l)} and f^{(o)}? Are these MLPs?

I also was confused in the experiments by the >= and <= on the reported numbers. For example, in table 2, the text describes the values displayed as log-likelihoods, in which case the ELBO represents a lower bound. However, in that case, why is the bolded value the *lowest* log-likelihood? That would be the worst model, not the best — does table 2 actually show negative log-likelihoods, then? In which case, though, the numbers from the ELBO should be upper bounds, and the >= should be <=. Looking at figure 4, it seems like visually the STCN and VRNN have very good reconstructions, but the STCN-dense has visual artifacts; this would correspond with the numbers in table 2 being log-likelihoods (not negative), in which case I am confused only by the choice of which model to bold.



UPDATE:

Thanks for the clarifications and edits. FWIW I still find the depiction of the architecture in Figure 2 to be incredibly misleading, as well as the decision to omit dependencies from the distributions p and q at the top of page 5, as well as the use in table 3 of "ELBO" to refer to a *negative* log likelihood.

---

> ### Author Response · Authors · 2018-11-17
> **Comments and Clarifications**
>
> *** Clarifications for figures and equations
> We apologize for the confusion. As the reviewer mentions the dilated convolutional stacks d_t^l has dependency reaching further and further back in time.
> In the original Fig. 2 we aimed to simplify the model details and show only a graphical model representation. The caption provides an explanation of the (updated) figure in the revised version. Moreover, the “Conv” equation (Eq. 2 in the revised version) is now a corrected to be a function of multiple time-steps, explicitly showing the hierarchy across time.
>
> ***Details of the inference and generative networks
> The difference between the prior and the approximate posterior, i.e., inference network are the respective input time-steps. The prior at time-step t is conditioned on all the input sequence until t-1, i.e., x_{1:t-1}. The inference network, on the other hand, is conditioned on the input until step t, i.e., x_{1:t}.
> At sampling time, we only use the prior. In other words, the prior sample z_t (conditioned on  x_{1:t-1}) is used to predict x_t. Here we follow the dynamic prior concept of Chung et al. (2015). During training of the model, the KL term in the objective encourages the prior to be predictive of the next step.
>
> *** f^{(l)} and f^{(o)} functions.
> f^{(l)} stands for neural network layers consisting of 1d convolution operations with filter size 1: Conv -> ReLu -> Conv -> ReLu which is then used to calculate mu and sigma of a Normal distribution.
>
> f^{(o)} corresponds the output layer of the model. Depending on the task we either use 1d Conv or Wavenet blocks. Network details are provided in the appendix of the revised paper.
>
> *** Clarification on MNIST results.
> This was indeed a typo. We report negative log-likelihood performance, measured by ELBO. We correct this in the revised version.
> In Fig. 4 (in the submitted version) we wanted to emphasize that STCN-dense can reconstruct the low-level details such as noisy pixels, which results in large improvement in the likelihood. We agree the STCN and VRNN provide smoothed and perceptually beautiful results. However, such enhancements lower the likelihood performance. Since the figure did not convey this clearly, we updated the figure in the revised version.
>
> ***References
> Junyoung Chung, Kyle Kastner, Laurent Dinh, Kratarth Goel, Aaron C Courville, and Yoshua Bengio. A recurrent latent variable model for sequential data. In Advances in neural information processing systems, pp. 2980–2988, 2015.

---

> ### Author Response · Authors · 2018-12-03
> **Response to the update**
>
> If it is advised by the reviewer, we would be glad to improve Figure 2. We aimed to visualize dense connections and highlight the difference between STCN and STCN-dense models in Figure 2 as a graphical model. Figure 5 (in appendix section) could be used as a replacement of Figure 2.
>
> “... decision to omit dependencies from the distributions p and q at the top of page 5...” this is because we don’t follow standard conditioning procedure. In other words, the top-most layer is only conditioned on d_t^L while the lower layers (l+1) depend on d_t^l and z_t^l.
>
> We will update Table 3 to the same convention used in other tables, i.e., NLL measured by ELBO.

---

### Official Review · AnonReviewer1 · 2018-11-05
**Ok paper with a reasonable -- though somewhat obvious -- approach to generative modeling of sequence data**

**Rating:** 6
**Confidence:** 5

**Review:**

This paper presents a generative sequence model based on the dilated CNN
popularized in models such as WaveNet. Inference is done via a hierarchical
variational approach based on the Variational Autoencoder (VAE). While VAE
approach has previously been applied to sequence modeling (I believe the
earliest being the VRNN of Chung et al (2015)), the innovation where is the
integration of a causal, dilated CNN in place of the more typical recurrent
neural network.

The potential advantages of the use of the CNN in place of
RNN is (1) faster training (through exploitation of parallel computing across
time-steps), and (2) potentially (arguably) better model performance. This
second point is argued from the empirical results shown in the
literature. The disadvantage of the CNN approach presented here is that
these models still need to generate one sample at a time and since they are
typically much deeper than the RNNs, sample generation can be quite a bit
slower.

Novelty / Impact: This paper takes an existing model architecture (the
causal, dilated CNN) and applies it in the context of a variational
approach to sequence modeling. It's not clear to me that there are any
significant challenges that the authors overcame in reaching the proposed
method. That said, it certainly useful for the community to know how the
model performs.

Writing: Overall the writing is fairly good though I felt that the model
description could be made more clear by some streamlining -- with a single
pass through the generative model, inference model and learning.

Experiments: The experiments demonstrate some evidence of the superiority
of this model structure over existing causal, RNN-based models. One point
that can be drawn from the results is that a dense architecture that uses multiple levels of the
latent variable hierarchy directly to compute the data likelihood is
quite effective. This observation doesn't really bear on the central message
of the paper regarding the use of causal, dilated CNNs.

The evidence lower-bound of the STCN-dense model on MNIST is so good (low)
that it is rather suspicious. There are many ways to get a deceptively good
result in this task, and I wonder if all due care what taken. In
particular, was the binarization of the MNIST training samples fixed in
advance (as is standard) or were they re-binarized throughout training?

Detailed comments:
- The authors state "In contrast to related architectures (e.g. (Gulrajani et
al, 2016; Sonderby et al. 2016)), the latent variables at the upper layers
capture information at long-range time scales" I believe that this is
incorrect in that the model proposed in at least Gulrajani et al also

- It also seems that there is an error in Figure 1 (left). I don't think
there should be an arrow between z^{2}_{t,q} and z^{1}_{t,p}. The presence
of this link implies that the prior at time t would depend -- through
higher layers -- on the observation at t. This would no longer be a prior
at that point. By extension you would also have a chain of dependencies
from future observations to past observations. It seems like this issue is
isolated to this figure as the equations and the model descriptions are
consistent with an interpretation of the model without this arrow (and
including an arrow between z^{2}_{t,p} and z^{1}_{t,p}.

- The term "kla" appears in table 1, but it seems that it is otherwise not
defined. I think this is the same term and meaning that appears in Goyal et
al. (2017), but it should obviously be defined here.

---

> ### Author Response · Authors · 2018-11-17
> **Comments and Clarifications**
>
> ***"significant challenges that the authors overcame in reaching the proposed method."
> The goal of our work was to design a modular extension to the vanilla TCN, while improving the modelling capacity via the introduction of hierarchical stochastic variables. In particular, we did not want to modify deterministic TCN layers (as is the case for Stochastic WaveNet, Lai et al., 2016) since this may limit scalability, flexibility and may limit the maximum receptive field size.
> These goals are motivated by findings from the initial phases of the project:
> 1) Initial attempts involved standard hierarchical latent variable models, none outperformed the VRNN baseline.
> 2) The precision-weighted update of approximate posterior, akin to LadderVAEs, significantly improved experimental results.
> 3) As can be seen from our empirical results, the increasing receptive field of TCNs provides different information context to different latent variables. This enables our architectures to more efficiently leverage the latent space and partially prevents latent space collapse issues highlighted in the literature (Dieng et al., 2018,  Zhao et al., 2017). The introduction of skip connections from every latent variable to the output layer directly in the STCN-dense variant seems to afford the network the most flexibility in terms of modelling different datasets (see p.8 & Tbl. 3 in the revised paper).
>
> ***  Effectiveness of TCN and densely connected latent variables
> Thanks for the interesting question. We agree that using multiple levels of the latent variables directly to make predictions is very effective. As we explain in the revised version of our submission, in STCN and STCN-dense models, the latent variables are provided with a different level of expressiveness. Hence, depending on the task and dataset, the model can focus on intermediate variables which have a different context. We think that this is an important aspect of our work, which can only be achieved by using the dilated CNNs. One can stack RNN cells similar to TCN blocks and use our densely connected latent space concept. In this scenario, the hierarchy would only be implicitly defined by the network architecture. However, since the receptive field size does not change throughout the hierarchy it is unclear whether the same effectiveness would be attained. Moreover, we note that combining our hierarchical stochastic variables with stacked LSTMs would inverse the effect on computational efficiency that we gain from the TCNs.
>
> ***“MNIST performance
> Yes, binarization of the MNIST is fixed in advance. We followed the procedure detailed in the Z-forcing paper closely. Naturally, we will release code and pre-processing scripts so that the results can be verified. Here is our experimental protocol:
> 1) We used the binarized MNIST dataset of Larochelle and Murray (2011). It was downloaded from http://www.cs.toronto.edu/~larocheh/public/datasets/binarized_mnist/binarized_mnist_train.amat
> 2) We trained all models without any further preprocessing or normalization. The first term of the ELBO, i.e., the reconstruction loss, is measured via binary cross-entropy.
> We provide an in-depth analysis in the revised version, showing that the STCN-dense architecture makes very precise probability predictions, also for pixel values close to character discontinuities. This provides very accurate modeling of edges and in consequence, gives very good likelihood performance. See (new) Figure 4 in the revised version.
>
> *** Clarifications
> We updated and clarified the Figure in the revised version. The generative model only relies on the prior. At sampling time, samples from the prior latent variables are used both in prediction of the observation and computation of the next layer’s latent variable. Therefore the generative model takes the input sequence until t-1, i.e., x_{1:t-1} in order to predict x_t.
> “The term "kla" appears in table 1, but it seems that it is otherwise not defined. I think this is the same term and meaning that appears in Goyal et al. (2017), but it should obviously be defined here.”
> Yes. It stands for annealing of the weight of KL loss term. We now clarified the language in tables and captions.
>
> ***References
> Lai, G., Li, B., Zheng, G., & Yang, Y. (2018). Stochastic WaveNet: A Generative Latent Variable Model for Sequential Data. arXiv preprint arXiv:1806.06116.
> Adji B Dieng, Yoon Kim, Alexander M Rush, and David M Blei. Avoiding latent variable collapse with generative skip models. arXiv preprint arXiv:1807.04863, 2018.
> Shengjia Zhao, Jiaming Song, and Stefano Ermon. Learning hierarchical features from generative models. arXiv preprint arXiv:1702.08396, 2017.
> Larochelle, Hugo, and Iain Murray. The neural autoregressive distribution estimator. Proceedings of the Fourteenth International Conference on Artificial Intelligence and Statistics. 2011.

---

> ### Author Response · Authors · 2018-12-14
> **Effectiveness of TCN and densely connected latent variables**
>
> To better understand if the experimental improvements shown in our paper only stem from the hierarchical latent space or whether the synergy between the dilated CNNs and latent variable hierarchy is important, we ran additional experiments (as suggested by R1). We replaced the deterministic TCN blocks with LSTM cells and kept the latent structure intact, dubbed RNNLadder. We used TIMIT and IAM-OnDB for speech and handwriting datasets. The log-likelihood performance measured by ELBO is provided below:
>
> =======================================================
>                                                                              TIMIT          IAM-OnDB
> =======================================================
>   25x256-LadderRNN (Normal)                         28207             1305
>   25x256-LadderRNN-dense (Normal)             27413             1278
> =======================================================
>   25x256-LadderRNN (GMM)                             24839             1381
>   25x256-LadderRNN-dense (GMM)                 26240             1377
> =======================================================
>   5x512-LadderRNN (Normal)                           49770             1299
>   5x512-LadderRNN-dense (Normal)               48612             1374
> =======================================================
>   5x512-LadderRNN (GMM)                               47179             1359
>   5x512-LadderRNN-dense (GMM)                   50113             1581
> =======================================================
>   25x256-STCN (Normal)                                    64913             1327
>   25x256-STCN-dense (Normal)                        70294             1729
> =======================================================
>   25x256-STCN (GMM)                                        69195             1339
>   25x256-STCN-dense (GMM)                            71386             1796
> =======================================================
>
> Models in the table have similar number of trainable parameters. The most direct translation of the the STCN architecture into an RNN counterpart has 25 stacked LSTM cells with 256 units each. Similar to STCN, we use 5 stochastic layers. Please note that stacking this many LSTM cells is unusual and resulted in instabilities during training. The performance is similar to vanilla RNNs. Hence, we didn’t observe a pattern of improvement with densely connected latent variables. The second RNNLadder configuration uses 5 stacked LSTM cells with 512 units and a one-to-one mapping with the stochastic layers.
>
> This experiments show that the modular structure of our latent variable design does allow for the usage of different building blocks. Even when attached to LSTM cells, it boosts the log-likelihood performance (see 5x512-LadderRNN), in particular when used with dense connections. However, the empirical results suggest that the densely connected latent hierarchy interacts particularly well with dilated CNNs. We believe this is due to the hierarchical nature in both sides of the architecture. On both datasets STCN models achieved the best performance and presented significant improvements with the dense connections. This supports our contribution of a latent variable hierarchy, which models different aspects of information from the input time-series.

---

### Author Response · Authors · 2018-11-17
**Updates in the revised paper**

We thank all reviewers for their constructive comments. Our work combines the computational advantages of temporal convolutional networks (TCN) with the representational power and robustness of stochastic latent spaces. Based on the reviewer’s feedback we have prepared an updated revision of the paper. Furthermore, we will respond to each review in a detailed manner below.

The most important changes in the revised version can be summarized as follows:
- We cleaned up the description of the background, method and improved the figures describing our model.
- We include an extensive discussion of related work as suggested by R3 and include direct comparisons to the state-of-the-art, where possible.
- During our experiments, we found that using separate \theta and \phi parameters for f^{l} is much more efficient, than to share the parameters of f^{l} (i.e., layers calculating mean and sigma of Normal distributions of the latent variables) for the prior and approximate posterior as suggested by Sønderby et al. (2016) and as was the case at submission time.
- With this change implemented, we re-ran experiments and updated the tables in the paper. On IAM-OnDB, Deepwriting,    TIMIT and MNIST we now report state-of-the-art log-likelihood results (even compared to additional models listed by R3). We also evaluate our model on the Blizzard dataset where only the Variational Bi-LSTM architecture is marginally better than STCN-dense (i.e., 17319 against 17128) but has access to future information.
- We include additional results on MNIST and provide insights why STCN-dense gives a large improvement in terms of reconstruction.
- We updated figures and equations throughout to improve clarity of presentation.

-----
Casper Kaae Sønderby, Tapani Raiko, Lars Maaløe, Søren Kaae Sønderby, and Ole Winther. Ladder variational autoencoders. In Advances in neural information processing systems, pp. 3738–3746, 2016.

---

### Meta-Review · Area_Chair1 · 2018-12-15

**Confidence:** 4
**Recommendation:** Accept (Poster)

**Metareview:**

The paper presents a generative model of sequences based on the VAE framework, where the generative model is given by CNN with causal and dilated connections.

Novelty of the method is limited; it mainly consists of bringing together the idea of causal and dilated convolutions and the VAE framework. However, knowing how well this performs is valuable the community.

The proposed method appears to have significant benefits, as shown in experiments. The result on MNIST is, however, so strong that it seems incorrect; more digging into this result, or sourcecode, would have been better.